# Barriers and facilitators to satisfaction with diabetes care: The perspectives of patients attending public diabetic clinics in Dar es Salaam, Tanzania

**Emmanuel Z. Chona**[1]*, **Lusajo F. Kayange**[1], **Masunga K. Iseselo**[2]

**1** Muhimbili University of Health and Allied Sciences, Dar es Salaam, Tanzania, **2** Department of Clinical Nursing, Muhimbili University of Health and Allied Sciences, Dar es Salaam, Tanzania

* emmanuelchona20@gmail.com

**Data Availability Statement:** All relevant data are within the paper and its Supporting information files.

## Abstract

### Background

The prevalence of diabetes has been increasing steadily over the past decade in low- and middle-income countries (LMICs) with about three-quarters of people living with the disease globally residing in these countries. Patient satisfaction can be used as a proxy measure of overall facility performance, and its use has been recommended for determining the quality of services provided by healthcare centres and organizations. This study aimed to explore barriers and facilitators towards satisfaction with diabetes care among patients attending public diabetic clinics in Dar es Salaam, Tanzania.

### Methods

A qualitative descriptive study was carried out among people with diabetes attending public diabetic clinics in Dar es Salaam, Tanzania. Using a purposeful sampling technique, 35 people with diabetes were interviewed from May 2023 to July 2023 with the principles of saturation guiding sample size determination. A semi-structured face-to-face interview guide was employed in data collection. The audio-recorded interviews were transcribed and analyzed using a conventional content analysis approach after translation. NVivo 12.0 computer software was employed to organize and code the data.

### Results

A total of 35 participants were enrolled in this study with a mean (±SD) age of 58.5 (±13.76) years. Four predominant themes and 12 categories were identified after data analysis including two barriers and two facilitators toward patients' satisfaction with diabetes care. Financial constraints and unfavourable clinic environments were identified as barriers. Furthermore, good provider-patient relationships and continuity of care emerged as facilitators.

**Funding:** This study was funded by Tanzania Diabetes Association in collaboration with the Muhimbili University of Health and Allied Sciences.

## Conclusion

Barriers and facilitators to patients' satisfaction with diabetes identified in this study are greatly determined by socio-economic and cultural conditions, highlighting the role of the healthcare delivery systems and allied stakeholders in regulatory and policy development to address the existing barriers and consolidate the proven facilitators.

## Introduction

The prevalence of diabetes has been increasing steadily over the past decade in low- and middle-income countries (LMICs) with about three-quarters of people living with the disease globally residing in these countries [1]. In the year 2021, the International Diabetes Federation (IDF) reported that about 537 million adults were living with diabetes globally. The burden of the disease is expected to increase to 643 million adults by 2030 and 783 million adults by 2045 with substantial socio-economic impact among the affected population [1]. By the year 2021, more than 24 million adults were living with diabetes in Africa and this number was predicted to increase to 55 million adults by the year 2045 [1]. Moreover, diabetes was responsible for 416,000 deaths in the African region in 2021. By the year 2022, the estimated prevalence of diabetes in adults was 10.3% (over 2.8 million) in Tanzania [2].

With challenges encountered by low- and middle-income countries in tackling diabetes, the rising prevalence and effects associated with the disease is a work-up call to improve health care and prioritize diabetes as a major health concern [3, 4]. One of the strategies to respond is by having well-equipped healthcare systems that are accessible with comprehensive human and non-human resources [4, 5]. Thus, studies on patient satisfaction have emerged as an increasingly crucial health outcome and are used to identify which aspects of the services need significant modification for ultimate quality improvement and credibility of the diabetes centres or hospitals [6]. In low- and middle-income countries, healthcare delivery systems have received many negative comments, both from their patients and the society at large [7, 8]. These negative comments depict the poor quality of services provided by the healthcare systems which range from, health workers' technical incompetence, and delays in service delivery to poor accessibility of the services [8, 9]. These negative comments have led to poor public utilization of healthcare facilities and services offered as they are no longer attractive to consumers [9].

Patient satisfaction refers to an individual patient's cognitive appraisal of the extent to which the health services offered met his/her subjective expectations [10, 11]. Patient satisfaction with health services is, therefore, a complex and multidimensional concept comprising of structure, process, and outcome of care. Hence, it can be used as a proxy measure of overall hospital performance, and its use has been recommended for evaluating the quality of care provided by healthcare organizations [12, 13]. Currently, most of the low- and middle-income countries (LMICs) are studying patient satisfaction with health care to understand the patient experiences with the quality of services offered at the health facilities and the nature of problems/difficulties faced by their patients at the hospitals [14]. Diabetes clinics with a high patient satisfaction score, for example, have been reported to have high credibility in comparison with clinics having a low patient satisfaction score [15, 16]. According to the findings in different parts of the world, patients who are satisfied with the health services offered are more likely to utilize health services in the future and comply with the treatment protocols [15, 17, 18].

Several barriers have been identified to influence the quality of healthcare delivery, particularly in the context of low- and middle-income countries [12]. Poor knowledge about diabetes

among patients, shortage of trained specialists to attend to patients, inadequate medication supply, and poor competence among primary care providers are among the barriers to the provision of quality treatments at the diabetes centres in these countries [14, 16]. Among the challenges that prevent most diabetes patients from accessing the clinics' services for diabetes management is dissatisfaction with diabetes care once they visit the clinics and because they are dissatisfied, it is most likely they will not return to the services in the future [19–21]. Also, despite the magnitude and burden of chronic diseases like diabetes in low- and middle-income countries, services for the diagnosis, and treatment, of diabetes are progressive satisfying the patients in some aspects of the care offered. This includes the availability of online information concerning self-management of diabetes disease through the websites of the hospitals that enables patients to gain insight into their disease conditions and make proper follow-up with medical treatments [22, 23].

In Tanzania, most of the diabetes health services are provided by public diabetes clinics established within hospitals in district and regional capitals. Patients from peripheral rural areas need to travel some distances to towns where they can access diabetes services and most of them seek diabetes care during emergencies, and without scheduled appointments [24]. Despite several studies done in the context of the current study to examine the extent of satisfaction with diabetes care, there is still inadequate information on patients' perspectives about potential barriers and facilitators toward satisfaction with the services provided. Therefore, this study aimed to explore barriers and facilitators towards satisfaction with diabetes care among patients attending public diabetic clinics in Dar es Salaam, Tanzania.

## Materials and methods

### Study design

A qualitative descriptive study design was employed to explore barriers and facilitators towards satisfaction with diabetes care among patients attending public diabetic clinics in Dar es Salaam, Tanzania. This design was employed because it allows an in-depth exploration of participants' experiences with potential barriers and facilitators to the delivery of satisfactory diabetes care at the facilities [25].

### Study setting

The study was carried out at Muhimbili National Hospital(MNH)-Mloganzila and the three municipal hospitals (Amana, Temeke, and Mwananyamala) in Dar es Salaam. Being a business capital city, Dar es Salaam attracts a lot of people to reside within it and that is why there is a high population compared to other cities in Tanzania. Moreover, the city has a national hospital and municipal hospitals with diabetic clinics where a large number of the affected population attends. MNH-Mloganzila is the National Referral Hospital with a 608-bed capacity and attends about 1,200 outpatients per day. The diabetic clinic is one of the clinics in the MNH-Mloganzila; it operates 5 days a week. About 60–90 patients are consulted by doctors on each clinic appointment day. Amana Hospital is located in the Ilala district and diabetes clinics are conducted twice per week with about 30 patients per day. Temeke Hospital is located in the Temeke district in Dar es Salaam. Diabetic clinics are conducted three times a week and about 40 patients attend per day. Mwananyamala Hospital, on the other hand, is located in the Kinondoni district. The diabetes clinics at Mwananyamala Hospital are conducted three times per week with about 20 patients attending the clinic per day.

## Study population and eligibility criteria

This study involved a population of diabetes patients aged 18 years and above attending diabetic clinics at selected public hospitals in Dar es Salaam, Tanzania. Critically ill patients were excluded from this study because they might not be comfortable participating in interviews due to potential life-threatening physiological conditions requiring critical care. Also, patients who had communication problems like hearing and speaking problems were excluded from the current study.

## Sampling procedure and sample size

A purposeful sampling technique was used to recruit the study participants. Records of patients were reviewed at the registry point to identify those who had attended the clinic on a particular appointment day. People with diabetes who were able and willing to provide their descriptions were purposively selected from those who met the inclusion criteria. The decision to use this sampling technique was based on the willingness of participants to share their ideas and views concerning barriers and facilitators towards satisfaction with diabetes care at their respective health facilities. A total of 35 participants were interviewed with sample size determination based on the principles of saturation as proposed by Malterud et al., [26], therefore; sampling was terminated when no new information was obtained from the study participants.

## Data collection tool

In this study, a semi-structured interview guide developed by authors based on reviewed literature [18, 27, 28], and modified to suit the specific objectives of the study was used to collect information from the study participants. The guide was employed to gather in-depth information about the perceived barriers and facilitators toward satisfaction with diabetes care. Open-ended questions were employed to encourage discussion and enable participants to express their perceived barriers and facilitators (See Supporting information). A background information sheet was used to collect sociodemographic information from the study participants. The interview guides were developed in English and then translated into Swahili, the national language to enhance better understanding by the study participants. Furthermore, it was pre-tested before the data collection was carried out to ensure the appropriateness of the questions.

## Data collection procedure

Recruitment and in-depth interviews were conducted between May 2023 to July 2023 by the principal investigators (EZC and LFK) of this study with the assistance of 4 trained research assistants who had completed 4 days of intensive training at the data collection site. Patients waiting for their consultation at the waiting area were approached by the research assistants and invited to participate. Participation in the study was voluntary with no monetary incentives. All participants provided written informed consent and completed a background information sheet to collect demographic information including age, gender, duration of diabetes in years, marital status, educational background, and participant's occupation. The interviews were carried out at hospital premises in rooms providing privacy and comfort to avoid external distractions. In each interview, the questions in the semi-structured interview guide were followed by several probes for clarification of the responses provided and exploration of more information from the participants. Data collection was conducted in Swahili (the national language), and all interviews were digitally audio-recorded. Data that were not easily captured by the audio recorder, such as non-verbal cues were written down as field notes in the notebook

by the research assistants and later considered for analysis. The interviews were conducted till the point of saturation. The duration of in-depth interviews lasted between 30 to 45 minutes.

## Data analysis

Data collection and analysis were conducted concurrently so that early insights could inform the focus of later in-depth interviews. This also enabled the principal investigators to gauge the richness of the participant's accounts, which informed decisions about sample size. The analysis was initiated by transcription of the audiotaped data by typing directly into the computer with the aid of the Microsoft Word program and then translating it into the English language. To gain a general impression and overall understanding of the qualitative data, several iterations of the transcripts with an open mind were done by the principal investigators. Further detailed review of the transcripts was done by the principal investigators to identify patterns and meaning units that were used to generate a coding scheme. Codes were reviewed for clarity and duplication and emergent themes were added to the coding scheme to ensure completeness. The transcribed and translated data sets were organized and managed using qualitative data analysis computer software NVivo 12.0. The conventional content analysis approach was used to enable a deeper understanding of the transcripts and formation of themes after several reading iterations as proposed by Hsieh & Shannon [29]. Categories were developed from actual phrases in the text segments and similar categories were linked to form themes. Relevant categories were supported by direct quotes from the in-depth interviews. To ensure the trustworthiness of qualitative findings, the authors held a meeting to discuss meanings emerging from the analysis outputs (codes, categories, and themes) according to the specific objectives of this study. Moreover, the first draft of findings for five patients who participated in the survey and gave consent to be contacted later on were back-translated into the Swahili language and shared with them individually. Their suggestions were obtained regarding the comprehensiveness of the findings and changes were incorporated accordingly.

## Ethical considerations

This study was conducted in accordance with the Declaration of Helsinki [30], and received ethical approval from the Institutional Review Board in the Directorate of Research and Publications of the Muhimbili University of Health and Allied Sciences with Ref. No. DA.282/298/01.C/1653. All administrative authorities from the respective health institutions permitted the data collection. Written informed consent to participate in the study was obtained from each participant before being interviewed. Anonymity and confidentiality of the participants' information were strictly maintained by removing any personal identifiers from the data, and the audio-recorded interview transcripts were kept in a password-protected electronic device accessible only to the principal investigators and the co-authors. Moreover, each potential participant was informed of his/her right to decline participation outright or to withdraw consent at any stage of the study without undesirable consequences.

## Findings

### Socio-demographic characteristics of study participants

A total of 35 participants were enrolled in this study. The mean (±SD) age of participants was 58.5 (±13.76) years with more than half of the participants in the age group of 46 to 65 years. In this study, 19 participants were male patients and only 2 participants had 21–30 years since being diagnosed with diabetes. Six participants were single, the other 7 were divorced, and only 5 had college/higher education (Table 1).

**Table 1. Socio-demographic characteristics of study participants.**

| Variables | Frequencies, n(%) |
|---|---|
| **Age of participant** | |
| 26–45 years | 6(17.2) |
| 46–65 years | 20(57.1) |
| 66–85 years | 9(25.7) |
| **Sex of participant** | |
| Male | 19(54.3) |
| Female | 16(45.7) |
| **Duration of diabetes since diagnosis** | |
| 1–10 years | 22(62.9) |
| 11–20 years | 11(31.4) |
| 21–30 years | 2(5.7) |
| **Participant's marital status** | |
| Single | 6(17.2) |
| Married | 11(31.4) |
| Widow/widower | 11(31.4) |
| Divorced | 7(20.0) |
| **Participant's level of education** | |
| No formal education | 5(14.3) |
| Primary education | 16(45.7) |
| Secondary education | 9(25.7) |
| College/higher education | 5(14.3) |
| **Participant's occupation** | |
| Formally employed | 6(17.2) |
| Self-employed | 8(22.9) |
| Unemployed | 16(45.7) |
| Retired | 5(14.3) |
| **Time since enrollment at the clinic** | |
| < 6 months | 5(14.3) |
| 6–12 months | 12(34.3) |
| > 12 months | 18(51.4) |
| **Average travel time from home to health facility by car** | |
| < 1 hour | 10(28.6) |
| 1–3 hours | 17(48.5) |
| > 3 hours | 8(22.9) |
| **Average duration of time spent at the facility per visit** | |
| < 1 hour | 6(17.2) |
| 1–3 hours | 12(34.3) |
| > 3 hours | 17(48.5) |

## Themes identified

Four predominant themes and 12 categories were identified after data analysis; two of the themes were barriers and two of them were facilitators to satisfaction with diabetes healthcare services provided at the public clinics. Financial constraints, and unfavourable clinic environments were identified as barriers. Furthermore, good provider-patient relationships and continuity of care emerged as facilitators (Table 2).

**Table 2. Summary of themes and categories.**

| Barriers | |
|---|---|
| **Themes** | **Categories** |
| **(i) Financial constraints** | Cost of medications |
| | Limited health insurance coverage |
| | Perceived lack of government's financial support |
| | Scarcity of medicines and glucometers |
| **(ii) Unfavorable clinic environments** | Unfavorable waiting areas |
| | Technological challenges |
| | Shortage of trained diabetes care specialists |
| **Facilitators** | |
| **Themes** | **Categories** |
| **(i) Good provider-patient relationships** | Shared decision-making culture and healthcare plan |
| | Comfort and support during care provision |
| | Effective information exchange |
| **(ii) Continuity of care** | Unfragmented healthcare delivery systems |
| | Availability of sustainability plan |

**Barriers.** *(i) Financial constraints.* Participants revealed financial burdens associated with different aspects of diabetes care. They reported facing high out-of-pocket payments to cover different services provided at the public diabetic clinics despite their limited financial security. The theme is explained under four categories; cost of medications, limited health insurance coverage, perceived lack of government financial support, and scarcity of medicines and glucometers.

*Cost of medications.* Most participants revealed the high costs associated with medications for the management of diabetes as recommended by healthcare providers. They expressed that they were using a huge amount of money to buy the medicines that could serve them until their next clinic appointment. The latter forced them to sell the little resources and assets they had at home to purchase enough medications.

"*As we are speaking today, I have sold my little house I had there in Kiluvya because I was heavily indebted and most of the expenses were attributed to these medicines I'm using. I'm using three kinds of medications and all of them are very expensive for someone like me to purchase because I solely depend on relatives to fund my treatments and most of them are just peasants there in Visiga*"

(Participant 5).

In addition, many participants reported having reduced capability to perform their usual activities due to physiological distress associated with the disease. Others suffered disability as the outcome of treatment options for complications attributed to diabetes like amputation secondary to septic diabetic foot ulcer. This resulted in a long-term loss in financial earnings with less money for purchasing medications prescribed for diabetes management.

"*It has been two years now without my leg, during those years I used to cultivate maize and vegetables to get food and earn little amount of money for catering to other basic needs. Currently, as you can see my son, I can't do all those activities, and thereof I depend on my children for everything including hospital expenditures*"

(Participant 29).

***Limited health insurance coverage***. Most participants lacked comprehensive health insurance cards to accommodate all diabetes-related services provided at the clinics. They revealed that most of the medications were excluded from the community health fund cards they had and thus they had to purchase them with out-of-pocket money at the hospital and out-of-hospital pharmacies.

"*With the health insurance card I had, I had to use an extra amount of money to buy some of the drugs because doctors here told me that my card couldn't afford all of them. Also, some of these drugs are sometimes only available at street private pharmacies and are very expensive to purchase. I wish to have those NHIF cards because I'm being told that those are at least comprehensive meaning that they can buy all the drugs I'm using currently*"

(Participant 13).

Some participants were forced to borrow money from family members for treatment, thus financially affecting the whole extended family. Some participants mentioned experiencing family tension about the financial burden of their disease due to constant expenditure associated with regular clinic attendance for seeking treatments.

"*My relatives are a little bit upset now because I routinely ask them for money to buy the medicines I'm being prescribed with. Some of my close relatives have developed arguments to discourage others who help me some amount of money for attending the clinics by claiming that, they will run bankrupt for treating an incurable disease*"

(Participant 21).

***Perceived lack of government's financial support***. Participants expressed their thoughts that the government doesn't consider them as a key population because they are not financially supported. With such perceived negligence from the government which was believed to be the main stakeholder in organization and ultimate delivery of healthcare services, the care offered at public diabetes clinics was challenged with financial constraints among the affected population. They claimed to face financial difficulties in meeting basic expenses associated with their treatments like consultation fees and access to medications.

"*The government don't offer any significant support to diabetic patients particularly financially to provide us with the capability to pursue the healthcare services. …….. This clinic for instance is owned by the government but can you imagine we have to pay consultation fees each day we come here to get the services*"

(Participant 32).

Other participants went far in their explanations claiming unequal treatment among populations with chronic illnesses as some groups were given special consideration. They pointed out reference populations who are given special attention and consideration in terms of their treatments and even social support.

"*In my point of view, the government has neglected us unlike those people with HIV/AIDS as you know that those people are being offered free medicines to take daily and are sometimes empowered with balanced diet at their treatment centres*"

(Participant 18).

***Scarcity of medicines and glucometers***. Participants expressed that; the hospital pharmacies lacked some of the medicines they were prescribed. The latter forced them to look for external pharmacies to purchase the medicines which was associated with higher costs compared to those incurred at hospital pharmacies.

"*I don't know if there is no outstanding supply of these medicines to these hospital pharmacies or if there are other issues behind because many times, I'm being directed to the external pharmacy to access some of the medicines. And keep in mind that in those outside private pharmacies, the prices for these medicines are higher than those of in-hospital pharmacies*"

(Participant 23).

The scarcity of glucose monitoring devices was highlighted by participants as they lacked enough random blood glucose strips for measuring their blood glucose at home. They claimed to receive few strips in comparison to actual needs every month until the next visit when they could attend the next visit.

"*The strips I'm getting on each visit are not enough for monitoring the blood sugar until the next clinic visit. So, there are days and situations when we inject insulin without measuring the blood sugar levels. . .. . .. they always tell us that the strips are very few for each to get the required actual number monthly considering the number of patients in need, So, they have to distribute few strips so that everyone can get*"

(Participant 15).

With the scarcity of glucose monitoring devices, participants expressed their thoughts that it was sometimes very challenging to adhere to care providers' medical advice due to the lack of such glucose monitoring devices.

"*It is very difficult sometimes to have the insulin injection regularly because I think you know what can happen with it when you inject blindly (without measuring the current blood sugar). All these are attributed to lack of enough strips for measuring the blood sugar*"

(Participant 20).

*(ii) Unfavorable clinic environments*. Unfavourable clinic environments were reported as a barrier to patients' satisfaction with diabetes services at the clinics. This theme is explained under the following categories; unfavourable waiting areas, technological challenges, and shortage of trained diabetes care specialists.

***Unfavorable waiting areas***. Participants revealed the presence of ugly waiting places at the clinics. Poor ventilation in the waiting areas was among the escalating concerns among diabetic people with diabetes at the clinics. Also. lack of enough sitting spaces in proportion to the actual number of patients attending the clinic was reported to annoy them and that reduced their chances of getting satisfied with the overall delivery of diabetes services at the clinics.

"*The waiting area here as you can see is not friendly at all, firstly, the number of benches is not sufficient to accommodate the actual population of patients waiting for services here. Secondly, this place is poorly ventilated to the extent that we have to sometimes get out there to ventilate especially when the queue is very long*"

(Participant 11).

Some participants claimed that it could at least get better if there were long wait times with well-designed waiting places. Having both unpleasant waiting places and long wait times because of higher patient loads and inadequate staffing to attend to the patients was very devastating to them.

"*I advise that they have to modify this waiting area by adding more benches for patients to sit down and putting air conditioners around. The latter will at least enable us to wait for a long time before consultations tireless. . . . . . . . . I have said because I actually know it is not easy to employ more healthcare providers urgently considering our country's economic constraints but modifying the waiting area first is very possible*"

(Participant 16).

***Technological challenges***. Most participants revealed their concerns with care delivery network systems, they claimed to experience long wait times due to the unavailability of the network during clinic hours. Due to the healthcare delivery systems in recent years being unprecedented by an increase in the use of biomedical network systems, there have been remarkable internet problems in the operation of these systems which distorts the speedy delivery of services at the clinics.

"*We acknowledge the presence of computerized care delivery systems, yes but the system sometimes is very boring because you can wait for 1–2 hours the network is not working and we have to wait for it without getting treatments*"

(Participant 1).

"*The issues of internet problems here are very common, they are very repetitive to the extent that we have to spend more time to get the treatments. Therefore, it interferes with other work routines because we are being late here despite coming earlier in the morning*"

(Participant 35).

Moreover, patients were apprehensive that having network problems was perceived to increase the workload at the facility given the shortage of healthcare providers. The latter was associated with many patients leaving the clinic untreated on a particular clinic day after foreseeing such problems due to fear of losing the whole day waiting for the internet to return.

"*Today, one of my friends has recently left untreated because of network problems. He promised to return tomorrow for the services because with such a network problem and high patient load, as you can see, he would attend very late to his works there in the market*"

(Participant 28).

***Shortage of trained diabetes care specialists***. Participants revealed the shortage of trained diabetes care specialists at the clinics, and hence it took a long time to accomplish all the services on a particular clinic appointment visit. They were subjected to long wait times before doctor's consultations due to the high patient load at the clinics and the lack of enough specialists to attend to the population on a particular clinic appointment day.

"*As you have said in any success, barriers are obvious. In my point of view concerning this aspect, the barriers are like a shortage of care providers which makes us delay getting the*

*services. I wish we could have more specialized doctors and consultation rooms, it could be better than this situation*"

(Participant 33).

Some participants were subjected to adverse health outcomes as a result of long wait times at the clinics.

"*Eeeeeh listen, we diabetic patients sometimes suffer from the long waiting time especially when the blood glucose levels are elevated or lowered below normal ranges, we feel very uncomfortable and you find such a situation on your clinic appointment day, can you imagine how you are going to suffer the consequences. . . . . . . . . During my clinic visit here in January, I fainted here while waiting for my queue to see the doctor and therefore, I was admitted to the ward that day I didn't go back home as I used to do*"

(Participant 9).

**Facilitators.** *(i) Good provider-patient relationships.* During the interviews, participants repeatedly commented positively on different aspects of healthcare providers' interactions with patients. They expressed satisfaction with the presence of a shared decision-making culture and healthcare plan, comfort and support during care provision, and effective information exchange.

***Shared decision-making culture and healthcare plan.*** The presence of the right to decision-making regarding treatments of diabetes made patients develop a culture of critical thinking with reflections on an individual basis after being informed of their conditions and management protocols by healthcare providers on each clinic appointment day. Participants expressed that they were involved to a greater extent in the decisions made on their treatments after having mutual agreements with healthcare providers.

"*In this clinic, we are usually empowered and encouraged to give out our views on the treatments we are getting during the consultations. The doctors here give us that ability for making decisions regarding our health regardless of the situation because they usually tell us that health services are delivered only upon consent from patients*"

(Participant 4).

In case of a lack of consensus between the healthcare providers and patients, other allied parties were invited with consent from the patient to explore better options for the betterment of the patient's health.

"*Yes, as you have said there are situations we encounter here. For instance, during my last visit here I was unable to cover all the expenses plus the new medications my doctor prescribed for the first time. I rejected them because I had no money to purchase them but the doctor emphasized their importance because my blood sugar was very high. Social workers were then summoned by the doctor and after a couple of discussions, I was offered the medications with promise to compensate the expense on my subsequent visits*"

(Participant 21).

Also, participants reported the presence of a shared healthcare plan as they were directly involved in developing goals for the management of their conditions and tangible strategies were set collaboratively to make sure that the goals were met. They acknowledged the plans formulated for the maintenance of blood glucose levels every month through dietary modifications and adherence to treatment regimens.

"*We usually set goals collaboratively for maintenance of blood sugar, the healthcare providers do emphasize adherence to medications and dietary counselling sessions are conducted so that we can have a meal plan for regulating the blood sugar levels. . . . . . . . the plans are usually monthly based because evaluation is done monthly upon each clinic attendance*"

(Participant 34).

Other participants went far in their expressions claiming that, the healthcare services offered at the clinics recently are quite different from what was delivered in the past few years where there was no freedom for patients to express their concerns concerning the establishment of a sustainability plan with healthcare providers. They perceived the treatments offered in the past few years as orders for patients to implement, unlike current treatments where they are free to give out their feedback concerning certain drug regimens in case of side-effects encountered and ultimate compliance and thereof make possible adjustments to such regimens.

"*I can say nowadays things have changed a lot, in those years when we were attending the clinic there in Upanga, we had no freedom like what we are having in these recent few years. I can tell you, in those years during consultations, doctors and nurses provided healthcare services in this clinic as orders to implement. We had no chances to opt for alternative drugs when something was wrong with the medications we had*"

(Participant 10).

***Comfort and support during care provision***. Many participants reported the supportive roles of healthcare providers in their course of treatment as they were encouraged to comply with a healthy diet and seek medical help whenever they desired to do so. Patients felt comfortable with the healthcare services they were getting at the clinics and they had that self-motivation in adhering to treatment plans scheduled.

"*By the day I moved from my local clinic at the dispensary and came here to Temeke, I felt a big difference between the two contexts. At our dispensary, we attended the clinic just to get prescriptions to buy the medicines. . .Aaaah I mean there was no room for getting time to sit with nurses and doctors and explaining to them what has been going on with you from the last visit, unlike here we get time to sit together with them and they support us so that we are motivated to continue with treatments*"

(Participant 31).

Moreover, they felt supported and comforted by healthcare providers in difficult situations they encountered during their course of treatment. They revealed that there were situations where they lost hope with the treatments they were getting at the clinics and they had a lot of unanswered questions concerning the ultimate prognosis of their illnesses. Healthcare providers supported them not only through pharmacological management but also through emotional and psychological support.

"*With the aspect of support from the experts here, I can guarantee you that there is good support from the nurses and doctors not only through giving us these medicines like what I have here but also, they get time to talk with us and explore us a lot……. Psychologically? Yes, there are days we come here with unusually elevated blood sugar to the extent that we are disappointed with taking the medications but the nurses and doctors do encourage us to continue with them by telling us that it's sometimes normal to have such ups and downs in the maintenance of blood sugar*"

(Participant 7).

***Effective information exchange***. Many participants revealed communication as one of the main facilitators to their satisfaction with diabetic services at the clinics. There was a free exchange of information among healthcare providers and patients all being directed to therapeutic purposes.

"*In the doctor's room, they usually let us explain our concerns and after those clarifications are made, they give us feedback according to their knowledge and experience because you know what? What they tell us is what they have studied for many years in the class and they faced with other patients in their daily routines here at the clinic. Therefore, in my point of view there is good communication between them and us patients like what I have said they listen to us carefully and respond to our questions and we are satisfied with that*"

(Participant 30).

Participants also expressed that; the communication was not limited only to clinic premises as they were able to contact healthcare providers even when they were at home during working hours. The latter enhanced a good therapeutic relationship with an effective flow of information to both parties (healthcare providers and patients) for better management of diabetes and prevention of possible complications related to the disease.

"*It was yesterday in the mid-day, I experienced a persistent headache after taking Metformin drugs, I think you know them………. after telling my daughter about the situation, she advised me to phone Dr…….. because I had the numbers in my mobile phone and my daughter also had them. So, I spoke with him through the phone and he urged me to attend the clinic today despite that it was not my clinic day*"

(Participant 2).

The availability of customer care mobile services enabled easier linkage of patients with their healthcare providers in situations of inevitable distant consultations.

"*Yes, there is this number we have been encouraged to call it in case we have anything we don't understand. It is a customer care mobile phone number for helping patients when they are far away from these hospital premises. Despite that it operates with normal call charges, it helps many of us and we are happy with the response they give us through the phone*"

(Participant 14).

*(ii) Continuity of care*. One of the most dominant themes in the current study was the continuity of care which is explained under the following categories; unfragmented healthcare delivery systems, and availability of sustainability plans.

***Unfragmented healthcare delivery systems***. Participants revealed the presence of an integrated healthcare delivery system at the clinics that provided continuity of care and a more efficient referral process. They expressed satisfaction with how the information was handled through the computer system in a way that their treatment process did not interfere with the loss of previous notes of their illnesses. It was easier for the healthcare providers to retrieve patients' information for evaluation of ongoing plans and establishment of plans.

"*The system nowadays is very organized despite some minor challenges, the information is fed and saved in the computers for future use. For example, with the system they use here, they can track back the trend of your blood sugar upon each visit and see how it goes because it has been stored there. The prescriptions are also uploaded online so that the pharmacists can access them and dispense the medications easily*"

(Participant 9).

Other participants went far in their expressions comparing the current healthcare delivery systems and healthcare delivery systems in past years which was associated with more paper works. They revealed and acknowledged the progress being made till nowadays when the system is computerized with less paperwork.

"*In the past years, the system was very poor because we had a lot of unnecessary movements from one room to another. . . . . . . . . I mean there was more paperwork and we used to trace the availability of medicines physically unlike today. For example, there were situations, when the doctor prescribed medications and directed you to go to the pharmacy to get the drugs, but once you reached the pharmacy you are being told that such drugs were not there, you see? So, you have to go back to the doctor and tell him/her that the prescribed drug is not there-. . . . . . . . .Nowadays, the doctor can visualize the available drugs through the network system they use and if the drug is not there, he/she changes it or he/she directs you where you can access the required drugs*"

(Participant 3).

***Availability of sustainability plan***. Participants expressed appreciation for healthcare providers' collaborative efforts with patients in the establishment of a sustainability plan through which it was easier for them to control their blood sugar levels and lead a quality life. They reported that healthcare providers do establish an open discussion with them concerning the treatments they are getting and develop cumulative and actionable healthcare plans.

"*. . . . . . .. Yes, we are being counselled here on diet and meal plans so that we can sustain the balance of blood sugar. As you can see there on that board, they have printed those learning materials which are easier for us to understand how things are supposed to be and we are being counselled each visit here on how we can maintain our health through dietary modification and compliance with medications. . . . . .the plans do vary of course depending on individual condition because each one has his/her unique challenges with this disease*"

(Participant 24).

Moreover, the availability of sustainable education programs made patients well-informed about their disease conditions and treatment modalities. They revealed how easier it was for them to adhere to treatment and management protocols because they were well informed on several self-management practices in case, they were far from hospital premises.

"*Every day here at 6 a.m., teaching sessions are being conducted by nurses and the topics are not the same each day. They usually establish and share a timetable for what will be taught on a particular day. For example, today we had a session on diabetic foot syndromes, so you can see we have been encouraged not to wear tight shoes to avoid such complications*"

(Participant 17).

## Discussion

Evidence from recent studies shows that comprehensive diabetes care requires the active engagement of patients in managing their condition to achieve and maintain optimal health which is greatly determined by their satisfaction with the care provided at the health facilities [11]. This study highlights important barriers and facilitators towards satisfaction with diabetes care as reported by patients attending public diabetic clinics. The findings of the present study uncovered several barriers and facilitators to satisfaction with diabetes care that are common across the clinics and can inform interventions that consider the logistical factors for assuagement of expressed barriers and consolidation of revealed facilitators.

In this study, financial constraints were identified as a dominant theme where participants reported experiencing higher costs of medications, limited health insurance coverage, perceived lack of government financial support, and scarcity of medicines and glucometers. Having limited health insurance among diabetes patients in the context of the current study necessitated the use of out-of-pocket expenditure to purchase medications and cover consultation fees. They claimed to lack government support in terms of financial empowerment to enhance their capability to cover the costs associated with services, they were getting from the public health facilities. Consistent financial instability was revealed in a study done in Nigeria among diabetes healthcare providers where physicians were reported to encourage the use of locally produced diabetes medicines and dietary management approaches for diabetes due to the limited capability of patients to purchase modern drugs [14]. As revealed in this study, patients were subjected to potential adverse outcomes associated with the intake of oral hypoglycemic drugs and insulin injections due to the scarcity of blood glucose monitoring devices before the self-medication practices, particularly at their home premises. These findings corroborate those of a study conducted among people with diabetes and patients with chronic diseases in Ghana and Kenya respectively [12, 31]. Also, the scarcity of medicines reported in this study has been reported in explorative studies conducted among people with diabetes in Nepal and Ghana [23, 32]. These findings have practical implications for the provision of diabetes services and call for multi-sectoral engagement in conducting regular and individualized assessments to explore patients' financial capability for early and proper linkage to social support groups [33, 34]. However, more studies are needed to provide contemporary insights into fundamental predictors of enrollment in health insurance schemes among the affected population.

Unfavourable clinic environments in terms of waiting places and lack of outstanding care delivery network systems have been highlighted as barriers to satisfaction with diabetes care by participants of this study. The issues of internet problems have been revealed to prolong the waiting times for consultation sessions of patients at the clinics. This highlights the need for more consistent and complete installation of sustainable care delivery systems to ensure that lessons learned can be of direct benefit in the management of diabetes in such resource-limited settings [35, 36]. The existence of unfavorable waiting areas compounded the unpleasant experiences faced by the patients at their respective facilities therefore minimizing the possibility of complete engagement of patients in various aspects of their care and predicting both

compliance and utilization of the facility services. Consistent findings of unfavourable facility environments have been elucidated by other relevant studies [15, 21, 37, 38]. A systematic and qualitative meta-synthesis study revealed significant service delivery network problems, particularly among low-middle-income countries where most of the diabetes health services are provided by public clinics established within hospitals in district and regional capitals [39]. The shortage of diabetes care specialists revealed in this study is similar to what was found in a multicenter cross-sectional study conducted in Northwest Ethiopia among diabetes patients where participants reported a shortage of trained endocrinologists and high patient load to be a setback towards their treatment satisfaction [40]. The findings are suggestive of the inevitable capacity building of public facilities through staffing improvements to accommodate the affected population in the respective catchment areas [7, 41, 42].

Good provider-patient relationships were highlighted by participants in the current study to facilitate satisfaction with diabetes care at the clinics. A shared decision-making culture and healthcare plan enhanced comfort and support to patients and ultimately pledged effective information exchange between healthcare providers and patients. These findings are consistent with several related studies conducted in different settings [13, 43–45]. For instance, a study conducted among rural African-American adults diagnosed with type diabetes mellitus revealed the presence of profound therapeutic provider-patient partnerships which encouraged them to comply with treatments and seek regular appointments [46]. Moreover, a sense of empowerment was secured through education-facilitated diabetes-specific patient-provider communication [46, 47]. This finding reiterates the importance of continual capacity building among healthcare providers with special emphasis on aspects of customer care to reduce patient turnover and subsequently enhance effective utilization of the available public health facilities.

In the context of the current study, continuity of care was delineated through unfragmented healthcare delivery systems and the availability of sustainability plans. Despite the limited availability of sustainable internet services, participants in the present study appreciated the presence of a well-organized computerized system at the clinics which was utilized to facilitate the storage of patients' information for future reference and the reduction of paperwork in handling the data. A recent study conducted among adults with type 2 diabetes mellitus revealed patients' satisfaction with shared decision-making and sustainability plans developed collaboratively with their healthcare providers [48]. Also, these findings are supported by previous studies that have shown that having a sustainability plan in chronic illness management was significantly associated with patients' satisfaction [17, 49, 50]. These findings have practical implications in the provision of patient-centred care and underscore a need for key stakeholders to create a linkage between health care systems and the community. This includes the development of community-based supplementary solutions outside of the traditional facility-based model to increase the availability of credible diabetes services [10, 49, 51]. The social networks could be adapted and manipulated to complement the health care system in the provision of appropriate messages and practice guidance to the affected population [52].

## Study limitations

As patients who had communication problems like hearing and speaking problems were excluded from the current study, this introduces a potential for information bias due to the lack of insights from such a group of people who are attending the clinics. In this study, only the perspectives of patients were explored regarding the potential barriers and facilitators towards satisfaction with diabetes care. It would be better to explore the healthcare workers' perspectives on the studied topic. Also, this study employed in-depth interviews as the only

data collection technique, it could be helpful to triangulate our findings by using other data collection methods, such as focused group discussions.

## Conclusion

Findings from the current study reveal significant barriers and facilitators to patients' satisfaction with diabetes care that are greatly determined by socio-economic and cultural conditions. This highlights the role of the healthcare delivery systems and allied stakeholders in regulatory and policy development to cater to unmet needs in the care of people with diabetes and the consolidation of proven facilitators. Moreover, the findings are strongly suggestive of the need to augment the existing social support strategies for people with diabetes that are considerate of individual patient contexts.

## Supporting information

**S1 Text. A guide for in-depth interview.**
(DOCX)

**S2 Text. Excerpts of transcripts.**
(DOCX)

## Acknowledgments

We are grateful to all participants for taking the time to participate in this study. Special appreciation goes to the research assistants of this study, for organizing, contacting, and keeping appointments with study participants.

## Author Contributions

**Conceptualization:** Emmanuel Z. Chona, Lusajo F. Kayange, Masunga K. Iseselo.

**Data curation:** Emmanuel Z. Chona, Lusajo F. Kayange.

**Formal analysis:** Emmanuel Z. Chona, Lusajo F. Kayange, Masunga K. Iseselo.

**Funding acquisition:** Emmanuel Z. Chona, Lusajo F. Kayange.

**Investigation:** Emmanuel Z. Chona, Lusajo F. Kayange.

**Methodology:** Emmanuel Z. Chona, Lusajo F. Kayange, Masunga K. Iseselo.

**Project administration:** Emmanuel Z. Chona, Lusajo F. Kayange.

**Resources:** Emmanuel Z. Chona, Lusajo F. Kayange.

**Software:** Emmanuel Z. Chona, Lusajo F. Kayange.

**Supervision:** Masunga K. Iseselo.

**Validation:** Emmanuel Z. Chona, Lusajo F. Kayange, Masunga K. Iseselo.

**Visualization:** Emmanuel Z. Chona, Lusajo F. Kayange.

**Writing – original draft:** Emmanuel Z. Chona, Lusajo F. Kayange.

**Writing – review & editing:** Emmanuel Z. Chona, Lusajo F. Kayange, Masunga K. Iseselo.

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
