## [Decision Letter · Decision Letter 0]

10 Mar 2024

PONE-D-24-00042Barriers and facilitators to satisfaction with diabetes care: the perspectives of patients attending public diabetic clinics in Dar es Salaam, TanzaniaPLOS ONE

Dear Dr. Chona,

Thank you for submitting your manuscript to PLOS ONE. After careful consideration, we feel that it has merit but does not fully meet PLOS ONE’s publication criteria as it currently stands. Therefore, we invite you to submit a revised version of the manuscript that addresses the points raised during the review process.

We look forward to receiving your revised manuscript.

Kind regards,

Edward Zimbudzi

Academic Editor

PLOS ONE

Journal Requirements:

"This study was funded by Tanzania Diabetes Association in collaboration with the Muhimbili University of Health and Allied Sciences"

3. We note that your Data Availability Statement is currently as follows:  All relevant data are within the manuscript and its Supporting Information files

Reviewers' comments:

Reviewer's Responses to Questions

**Comments to the Author**

1. Is the manuscript technically sound, and do the data support the conclusions?

Reviewer #1: Yes

Reviewer #2: Yes

2. Has the statistical analysis been performed appropriately and rigorously? 

Reviewer #1: N/A

Reviewer #2: N/A

3. Have the authors made all data underlying the findings in their manuscript fully available?

Reviewer #1: Yes

Reviewer #2: Yes

4. Is the manuscript presented in an intelligible fashion and written in standard English?

Reviewer #1: Yes

Reviewer #2: Yes

5. Review Comments to the Author

Reviewer #1: Review comments

Generally

1. This is a very important paper on patient safety with diabetic care. However, the manuscript requires extensive English language editing among other issues observed.

2. The participants are better referred to as “people with diabetes” rather than “diabetic patients”

Abstract

3. Page 2 line 25- please change “3 fourths” to 3 quarters also page 4 line 60

4. Include brief findings on demographics.

Introduction

5. This section needs to be revised. The authors just talked about challenges without highlighting patient satisfaction. There should be at least a paragraph explaining patient satisfaction then attempt to link it with service delivery. The challenges are well highlighted but the authors fail to explain the gap in the information that is already known. THIS IS THE JUSTIFICATION OF THIS STUDY.

6. Page 4 line 75- change ‘ultimately’ to ‘ultimate”

7. Page 5 line 89- change ‘won’t’ to would not

Materials and methods

8. This section can be shorter. May the authors exclude elaborate descriptions about the cities. The actual settings and the justification for choosing them will suffice.

9. Page 6 line 118- change the tense to past tense.

10. Under “study population and eligibility”- exclusions are made based on compromised autonomy. This applies to critically ill patients. Excluding people with hearing and speaking problems is not justified as there can always be strategies to enhance communication without compromising their autonomy. There is a risk of missing out on important information based on a criterion that can be addressed. Think information bias!!!

11. Page 7 line 43- “information- rich” is consistent with key-informant interviews. Did the researchers do these? If not, they must rephrase this sentence.

12. Page 7 line 146- write ie in full.

13. Page 8- “under data collection tool” please describe the structure of the interview. From the title, it seems the themes reported were predetermined not really discerned/emerged/identified. However, this is still acceptable and should be reported as such.

14. Page 9- under ’data collection procedure’- line 174. Saturation is achieved once in a study, not at many points during data collection as the authors are implying. Can this be addressed appropriately?

15. Page 10 under ‘ethical consideration’- please mention the Declaration of Helsinki

Results

16. This section will be better referred to as “Findings” rather than “Results”

17. There is no need for elaborate narrations of what we can already see in the table. Just mention the highlights such as sample size, gender composition and mean age. The reader can refer to the table for the rest of the information.

18. Page 13 line one – consider “identified” instead of “discerned”

19. A lot of revision is required in this section. The themes needs to be reworked. Under barriers, the second theme “limited accessibility……..” is a continuation of “financial constraints” except for the first category that can fit under “unfavorable clinic environments”. The category ‘scarcity of medicines’ under “limited accessibility…………” is still part of financial constraints. Costs of diabetes go beyond medications. There is a huge cost of consumables like glucose strips and glucometers. These should be mentioned under financial constraints as well.

20. Page 16 from line 266-273. There is reference to “psychosocial issues” but they are not mentioned anywhere else in the manuscript. They are a significant factor in diabetes care especially in resource limited settings.

21. Page 17 line 282- change “enhance ” to “provide”

22. Page 18 line 316- See comment number 19

23. Page 19 line 333- see comment number 19

24. Page 20-lines 357 to 359- rephrase

25. Page 21 line 365- the category can be better presented as “technological challenges”

26. Page 22 line 405- rephrase page 23 line 415- change “comfortability” to “comfortable”

27. Page 23 lines426 to 428- rephrase

28. Page 24 line 454- remove names of people

29. Page 27 line 515- the category “Shared healthcare plan” is the same as “shared decision making culture” on page 22 from line 392. The 2 can be combined

Discussion

30. This section will also require extensive revision after reworking the themes.

31. Replace “Discussions” with “Discussion”

32. Page 29 line 549- explain “assuagement” or remove

33. Page 32- lines 609-612- the authors mentioned internet connectivity vas a barrier under findings but is now being referred to as a facilitator. Please clarify

Limitations

34. Line 622- the weakness mentioned looks like a strength of the study- THIS WAS A MULTICENTRE STUDY !!!!!!

35. We can not mention under representation and lack of generalizability in qualitative research. The findings are not meant to be generalized.

Conclusion

36. Lines 628 to 632- too long a sentence- rephrase

Acknowledgements

37. Remove names of people. It is identifying information.

Reviewer #2: Thank you for providing the opportunity to review the manuscript entitled “Barriers and facilitators to satisfaction with diabetes care: the perspectives of patients attending public diabetic clinics in Dar es Salaam, Tanzania”. The manuscript has comprehensively explored barriers and facilitators to the satisfaction of patients with DM on the diabetic care they are receiving in Dar es Salaam, Tanzania. Please find my comments on the manuscript as follows.

- Could you please mention the specific type of qualitative descriptive study design used in the study?

- The perspectives of critically ill patients and those with hearing and speaking abnormalities regarding the barriers and facilitators might be different from the other DM patients and important to note in this study. How do you justify their exclusion?

- Please, cite references of literature reviewed for customization of the IDI tool.

- The submitted IDI tool also does not reflect probing questions. Please include some probing questions used during the interview.

- Given the interview was conducted inside the hospital premises, while patients waiting for consultation, how do you see the potential bias that can be introduced? As the interview was about their satisfaction and barriers/facilitators, do you think the patients provide accurate information while they are there in the hospital?

- How was the trustworthiness (rigor) of the study maintained? Please state this in detail in the manuscript.

- Table 1 – please include %ge along with the absolute numbers (frequencies).

- “The current study aimed to explore barriers and facilitators towards satisfaction with diabetes care among patients attending public diabetic clinics in Dar es Salaam, Tanzania.” – this is repetitive. Please delete it from the discussion.

- In the discussions: The reasons for having limited health insurance need to be explored more in this study. Is it related to the disapproval of the insurance companies to provide insurance for patients with chronic diseases like DM? The implication of this finding has not been provided very well. Please revise it and provide public health and clinical implications of the finding.

- The scarcity of medicines and trained health workforce are not discussed well in the discussion. Particularly, try to explain the health workforce load and burden in relation to the barriers to satisfaction.

- Please include biases that can be introduced in this study as a limitation.

- It would have been also better if the healthcare workers' perspective had been captured. Please also include this as a limitation of the study.

6. PLOS authors have the option to publish the peer review history of their article (what does this mean?). If published, this will include your full peer review and any attached files.

Reviewer #1: No

Reviewer #2: **Yes: **Nebiyu Dereje

---

## [Author Response · Author response to Decision Letter 0]

19 Mar 2024

March 19, 2024

To

Editor-in-Chief

PLOS ONE Journal

 Dear Sir/Madam,

On behalf of my co-authors, I am grateful for the valuable and constructive review of our manuscript PONE-D-24-00042 entitled “Barriers and facilitators to satisfaction with diabetes care: the perspectives of patients attending public diabetic clinics in Dar es Salaam, Tanzania”. We have revised the manuscript according to editor and reviewers’ comments and made clarifications where required. The specific response to editor and reviewers’ comments is elaborated in details within the matrix response sheet below after this letter.

Also, we have amended the statement concerning funders role for this study as requested by the editor and currently reads as; “This study was funded by Tanzania Diabetes Association in collaboration with the Muhimbili University of Health and Allied Sciences. The funders had no role in study design, data collection and analysis, decision to publish, or preparation of the manuscript”.

Yours Sincerely,

……………….

Emmanuel Z. Chona (Corresponding Author)

E-mail: emmanuelchona20@gmail.com

Matrix response sheet on academic editor and reviewers’ comments to manuscript PONE-D-24-00042 entitled “Barriers and facilitators to satisfaction with diabetes care: the perspectives of patients attending public diabetic clinics in Dar es Salaam, Tanzania”

Editor’s comments

Response from author

Please ensure that your manuscript meets PLOS ONE’s style requirements, including those for file naming

The manuscript has been thoroughly checked to ensure that it follows the PLOS ONE’s author guidelines and meets the style requirements, including names of the uploaded documents.

Please see the revised manuscript and other documents uploaded.

Thank you for stating the following financial disclosure: “This study was funded by Tanzania Diabetes Association in collaboration with the Muhimbili University of Health and Allied Sciences”

Please state what role the funders took in the study. If the funders had no role, please state: “The funders had no role in study design, data collection and analysis, decision to publish, or preparation of the manuscript."" 

If this statement is not correct, you must amend it as needed. 

Since the funders were not involved with study procedures, the financial disclosure statement had been revised to make it comprehensive and currently reads as follows: “This study was funded by Tanzania Diabetes Association in collaboration with the Muhimbili University of Health and Allied Sciences. The funders had no role in study design, data collection and analysis, decision to publish, or preparation of the manuscript”.

Please see the rebuttal letter for resubmission of this manuscript.

We note that your Data Availability Statement is currently as follows:  All relevant data are within the manuscript and its Supporting Information files

Please confirm at this time whether or not your submission contains all raw data required to replicate the results of your study. Authors must share the “minimal data set” for their submission. PLOS defines the minimal data set to consist of the data required to replicate all study findings reported in the article, as well as related metadata and methods

With this submission, we have uploaded minimal data set (Excerpts of transcripts) required to replicate the results of our study as supporting information file.

Please see the excerpts of transcripts uploaded as supporting information file.

When completing the data availability statement of the submission form, you indicated that you will make your data available on acceptance. We strongly recommend all authors decide on a data sharing plan before acceptance, as the process can be lengthy and hold up publication timelines. Please note that, though access restrictions are acceptable now, your entire data will need to be made freely accessible if your manuscript is accepted for publication. This policy applies to all data except where public deposition would breach compliance with the protocol approved by your research ethics board. If you are unable to adhere to our open data policy, please kindly revise your statement to explain your reasoning and we will seek the editor's input on an exemption. Please be assured that, once you have provided your new statement, the assessment of your exemption will not hold up the peer review process.

Apart from sharing minimal data set (Excerpts of transcripts) required to replicate the results of our study as supporting information file, all other data pertaining to this study are available from the corresponding author without restrictions.

Reviewer’s comments

Response from author

Reviewer 1

Generally

This is a very important paper on patient safety with diabetic care. However, the manuscript requires extensive English language editing among other issues observed.

The revised manuscript has been thoroughly copyedited by an expert university faculty competent in English with a strong scientific background in the field of medical research to make it clear and unambiguous.

Please see the revised manuscript.

The participants are better referred to as “people with diabetes” rather than “diabetic patients”

The manuscript have been revised and the phrase “people with diabetes” has been preferably used instead of “diabetic patients” except for few phrases particularly quotations in the results section stated by participants in their narrations and those used in titles of references.

Please see the revised manuscript.

Abstract

Page 2 line 25- please change “3 fourths” to 3 quarters also page 4 line 60

“Three-quarters” in the revised manuscript have replaced the word “three-fourths”.

Please see the revised manuscript (page 2 line 25 and page 4 line 60).

Include brief findings on demographics.

A statement concerning demographic characteristics of participants have been added in the results section of the abstract.

Please see the revised manuscript (page 2 line 40).

Introduction

This section needs to be revised. The authors just talked about challenges without highlighting patient satisfaction. There should be at least a paragraph explaining patient satisfaction then attempt to link it with service delivery. The challenges are well highlighted but the authors fail to explain the gap in the information that is already known. THIS IS THE JUSTIFICATION OF THIS STUDY.

We have revised the section with addition of a paragraph explaining patient satisfaction and how it relates with service delivery. Moreover, a statement describing a gap in the information that is already known have been added in the last paragraph of the section to justify the current study. 

Please see the revised manuscript (page 5 line 82-94 and page 6 line 111-114).

Page 4 line 75- change ‘ultimately’ to ‘ultimate”

The word “ultimately” had been changed to “ultimate”

Please see the revised manuscript (page 4 line 75).

Page 5 line 89- change ‘won’t’ to would not

The word “won’t” had been changed to “would not”

Please see the revised manuscript (page 5 line 102).

Materials and methods

This section can be shorter. May the authors exclude elaborate descriptions about the cities. The actual settings and the justification for choosing them will suffice.

The section has been critically revised and elaborate descriptions about the cities in the study setting have been removed to make it shorter.

Please see the revised manuscript (page 7 line 126-139).

Page 6 line 118- change the tense to past tense.

The tense in the statement was changed to past tense. However, the full statement was removed during revision of the section to make it shorter as per above comment (comment number 8).

Please see the revised manuscript (page 7 line 126-139).

Under “study population and eligibility”- exclusions are made based on compromised autonomy. This applies to critically ill patients. Excluding people with hearing and speaking problems is not justified as there can always be strategies to enhance communication without compromising their autonomy. There is a risk of missing out on important information based on a criterion that can be addressed. Think information bias!!!

We acknowledge the comment, and the exclusion of such people has been added as a potential source of information bias in the study limitation section.

Please see the revised manuscript (page 33 line 621-623).

Page 7 line 143- “information- rich” is consistent with key-informant interviews. Did the researchers do these? If not, they must rephrase this sentence.

Since we did not do key-informant interviews in our study, the phrase have been revised to remove ambiguity.

Please see the revised manuscript (page 8 line 151-154).

Page 7 line 146- write ie in full.

Since the abbreviation i.e. is a Latin word, we have revised the sentence with the use of more appropriate English word rather than writing the long form of a Latin word inside an English statement. 

Please see the revised manuscript (page 8 line 155).

Page 8- “under data collection tool” please describe the structure of the interview. From the title, it seems the themes reported were predetermined not really discerned/emerged/identified. However, this is still acceptable and should be reported as such.

The description on structure of the interview has been added in the section. However, elaborate descriptions on the structure of interview and other interview procedures is presented under the “Data collection procedures” below after the “Data collection tool” part.

Please see the revised manuscript (page 8-9 line 160-163 and page 9 line 169-184).

Page 9- under ’data collection procedure’- line 174. Saturation is achieved once in a study, not at many points during data collection as the authors are implying. Can this be addressed appropriately?

We acknowledge the comment; the phrase with such implication has been revised to convey the intended and factual implication concerning data saturation.

Please see the revised manuscript (page 9 line 183).

Page 10 under ‘ethical consideration’- please mention the Declaration of Helsinki

The Declaration of Helsinki has been included in the “ethical considerations” part for the study methodology.

Please see the revised manuscript (page 11 line 209-211).

Results

This section will be better referred to as “Findings” rather than “Results”

The section heading has been renamed as “Findings”.

Please see the revised manuscript (page 12 line 220).

There is no need for elaborate narrations of what we can already see in the table. Just mention the highlights such as sample size, gender composition and mean age. The reader can refer to the table for the rest of the information.

The descriptions of socio-demographics characteristics findings have been revised to highlight only few variables as the readers can refer to the table for detailed information on other variables.

Please see the revised manuscript (page 12 line 222-226).

Page 13 line one – consider “identified” instead of “discerned”

The word “discerned” has been changed to “identified”.

Please see the revised manuscript (page 14 line 228).

A lot of revision is required in this section. The themes needs to be reworked. Under barriers, the second theme “limited accessibility……..” is a continuation of “financial constraints” except for the first category that can fit under “unfavorable clinic environments”. The category ‘scarcity of medicines’ under “limited accessibility…………” is still part of financial constraints. Costs of diabetes go beyond medications. There is a huge cost of consumables like glucose strips and glucometers. These should be mentioned under financial constraints as well.

The categories under second theme “limited accessibility to comprehensive care centres” has been regrouped into the first theme (financial constraints) and one category (lack of trained diabetes care specialists) has been moved to the “unfavorable clinic environments” theme.

Please see the revised manuscript (page 15).

Page 16 from line 266-273. There is reference to “psychosocial issues” but they are not mentioned anywhere else in the manuscript. They are a significant factor in diabetes care especially in resource limited settings.

In this particular study, we explored the barriers and facilitators towards satisfaction with diabetes care. Participants expressed the barriers like “limited health insurance coverage” as it was presented in page 16 from line 266-273 of the original submission manuscript. Hence, the referenced participant and some other expressed how they depended on other family members to fund their treatments because they had no health insurance cards. The effect of depending to other family members is what probably brought psychosocial issues but it was not significant in our study.

Page 17 line 282- change “enhance ” to “provide”

The word “enhance” has been changed to “provide” in the revised manuscript.

Please see the revised manuscript (page 18 line 292).

Page 18 line 316- See comment number 19

The categories have been revised and regrouped as per comment number 19.

Please see the revised manuscript (page 15).

Page 19 line 333- see comment number 19

The categories have been revised and regrouped as per comment number 19.

Please see the revised manuscript (page 15).

Page 20-lines 357 to 359- rephrase

The statement has been rephrased in the revised manuscript as required.

Please see the revised manuscript (page 20 line 342-343).

Page 21 line 365- the category can be better presented as “technological challenges”

The category has been presented as “technological challenges” in the revised manuscript.

Please see the revised manuscript (page 20 line 349).

Page 22 line 405- rephrase page 23 line 415- change “comfortability” to “comfortable”

The statement has been rephrased and the word “comfortability” has been changed to “comfortable” in the revised manuscript.

Please see the revised manuscript (page 25 line 440).

Page 23 lines 426 to 428- rephrase

The statement has been rephrased in the revised manuscript as required.

Please see the revised manuscript (page 25 line 451-453).

Page 24 line 454- remove names of people

The names of people have been removed in the revised manuscript.

Please see the revised manuscript (page 26 line 479).

Page 27 line 515- the category “Shared healthcare plan” is the same as “shared decision making culture” on page 22 from line 392. The 2 can be combined

The two categories have been combined in the revised manuscript to produce a category named as “shared decision-making culture and healthcare plan”.

Please see the revised manuscript (page 15).

Discussion

This section will also require extensive revision after reworking the themes.

The section has been extensively revised after reworking the themes.

Please see the revised manuscript (page 30-33).

Replace “Discussions” with “Discussion”

The term “discussions” has been replaced with the word “discussion”.

Please see the revised manuscript (page 30 line 541).

Page 29 line 549- explain “assuagement” or remove

The word “assuagement” as used in the sentence meant to mitigate or alleviate the identified barriers in the provision of diabetes care at the clinics.

Page 32- lines 609-612- the authors mentioned internet connectivity vas a barrier under findings but is now being referred to as a facilitator. Please clarify

The barrier reported in our findings is “lack of outstanding care delivery network systems” (has been renamed to “technological challenges”) due to repetitive internet problems during working hours. The facilitator reported is “unfragmented healthcare delivery systems” due to availability of computerized system (biomedical computer systems) utilized to facilitate delivery of services at the clinics. Hence, despite the limited availability of sustainable internet services (barrier), participants in the present study appreciated the presence of biomedical computer systems (facilitator) at the clinics that was utilized to facilitate the storage of patients’ information for future reference and the redu

---

## [Decision Letter · Decision Letter 1]

2 Apr 2024

PONE-D-24-00042R1Barriers and facilitators to satisfaction with diabetes care: the perspectives of patients attending public diabetic clinics in Dar es Salaam, TanzaniaPLOS ONE

Dear Dr. Chona,

Thank you for submitting your manuscript to PLOS ONE. After careful consideration, we feel that it has merit but does not fully meet PLOS ONE’s publication criteria as it currently stands. Therefore, we invite you to submit a revised version of the manuscript that addresses a few points raised during the review process. 

We look forward to receiving your revised manuscript.

Kind regards,

Edward Zimbudzi

Academic Editor

PLOS ONE

Journal Requirements:

Reviewers' comments:

Reviewer's Responses to Questions

**Comments to the Author**

1. If the authors have adequately addressed your comments raised in a previous round of review and you feel that this manuscript is now acceptable for publication, you may indicate that here to bypass the “Comments to the Author” section, enter your conflict of interest statement in the “Confidential to Editor” section, and submit your "Accept" recommendation.

Reviewer #1: (No Response)

Reviewer #2: All comments have been addressed

2. Is the manuscript technically sound, and do the data support the conclusions?

Reviewer #1: Yes

Reviewer #2: Yes

3. Has the statistical analysis been performed appropriately and rigorously? 

Reviewer #1: (No Response)

Reviewer #2: N/A

4. Have the authors made all data underlying the findings in their manuscript fully available?

Reviewer #1: Yes

Reviewer #2: Yes

5. Is the manuscript presented in an intelligible fashion and written in standard English?

Reviewer #1: Yes

Reviewer #2: Yes

6. Review Comments to the Author

Reviewer #1: Thank you for addressing all comments. However, minor revision is still required.

1. Please make sure to replace "diabetic patients" with 'people with diabetes'. Some statements have not been changed eg lines 33 and 563

2. Page 7 line 120- The study was a qualitative descriptive study and not a phenomenological one. Please remove" phenomenological approach"

Reviewer #2: Congratulations to the Authors. All of my prior comments have been addressed in the revised manuscript.

7. PLOS authors have the option to publish the peer review history of their article (what does this mean?). If published, this will include your full peer review and any attached files.

Reviewer #1: No

Reviewer #2: **Yes: **Nebiyu Dereje

---

## [Author Response · Author response to Decision Letter 1]

2 Apr 2024

April 02, 2024

To

Editor-in-Chief

PLOS ONE Journal

 Dear Sir/Madam,

On behalf of my co-authors, I am grateful for the valuable and constructive review of our manuscript PONE-D-24-00042R1 entitled “Barriers and facilitators to satisfaction with diabetes care: the perspectives of patients attending public diabetic clinics in Dar es Salaam, Tanzania”. We have revised the manuscript according to reviewer’s comments. The specific response to reviewer’s comments is elaborated in details within the matrix response sheet below after this letter.

Yours Sincerely,

……………….

Emmanuel Z. Chona (Corresponding Author)

E-mail: emmanuelchona20@gmail.com

Matrix response sheet on academic editor and reviewers’ comments to manuscript PONE-D-24-00042R1 entitled “Barriers and facilitators to satisfaction with diabetes care: the perspectives of patients attending public diabetic clinics in Dar es Salaam, Tanzania”

Reviewer’s comments

Response from author

Please make sure to replace “diabetic patients” with “people with diabetes”. Some statements have not been changed eg lines 33 and 563

We have revised the manuscript and replaced the phrases “diabetic patients” with “people with diabetes”.

Please see the revised manuscript (page 2 line 32, page 30 line 563, and page 34 line 633). 

Page 7 line 120 – The study was a qualitative descriptive study and not a phenomenological one. Please remove “phenomenological approach”

The phrase “phenomenological approach” has been removed from the revised manuscript.

Please see the revised manuscript (page 7 line 120-122).

---

## [Decision Letter · Decision Letter 2]

11 Apr 2024

Barriers and facilitators to satisfaction with diabetes care: the perspectives of patients attending public diabetic clinics in Dar es Salaam, Tanzania

PONE-D-24-00042R2

Dear Dr. Chona,

We’re pleased to inform you that your manuscript has been judged scientifically suitable for publication and will be formally accepted for publication once it meets all outstanding technical requirements.

Kind regards,

Edward Zimbudzi

Academic Editor

PLOS ONE

Additional Editor Comments (optional):

Reviewers' comments:

Reviewer's Responses to Questions

**Comments to the Author**

1. If the authors have adequately addressed your comments raised in a previous round of review and you feel that this manuscript is now acceptable for publication, you may indicate that here to bypass the “Comments to the Author” section, enter your conflict of interest statement in the “Confidential to Editor” section, and submit your "Accept" recommendation.

Reviewer #1: All comments have been addressed

2. Is the manuscript technically sound, and do the data support the conclusions?

Reviewer #1: Yes

3. Has the statistical analysis been performed appropriately and rigorously? 

Reviewer #1: N/A

4. Have the authors made all data underlying the findings in their manuscript fully available?

Reviewer #1: Yes

5. Is the manuscript presented in an intelligible fashion and written in standard English?

Reviewer #1: Yes

6. Review Comments to the Author

Reviewer #1: (No Response)

7. PLOS authors have the option to publish the peer review history of their article (what does this mean?). If published, this will include your full peer review and any attached files.

Reviewer #1: No
